

# Simulated seasonal impact on middle atmospheric ozone from high-energy electron precipitation related to pulsating aurorae

Pekka T. Verronen[1,2], Antti Kero[1], Noora Partamies[3,4], Monika E. Szeląg[2], Shin-Ichiro Oyama[5], Yoshizumi Miyoshi[5], and Esa Turunen[1]

[1]Sodankylä Geophysical Observatory, University of Oulu, Finland
[2]Space and Earth Observation Centre, Finnish Meteorological Institute, Finland
[3]The University Centre in Svalbard, Norway
[4]Birkeland Centre of Space Science, Norway
[5]Institute for Space-Earth Environmental Research, Nagoya University, Japan

**Correspondence:** P. T. Verronen (pekka.verronen@oulu.fi)

**Abstract.** Recent simulation studies have provided evidence that pulsating aurorae (PsA) associated with high-energy electron precipitation is having a clear local impact on ozone chemistry in the polar middle mesosphere. However, it is not clear if PsA are frequent enough to cause longer-term effects of measurable magnitude. There is also an open question of the relative contribution of PsA-related energetic electron precipitation (PsA-EEP) to the total atmospheric forcing by solar energetic particle precipitation (EPP). Here we investigate the PsA-EEP impact on stratospheric and mesospheric odd hydrogen, odd nitrogen, and ozone concentrations. We make use of the Whole Atmosphere Community Climate Model and recent understanding on PsA frequency, latitudinal and magnetic local time extent, and energy-flux spectra. Analysing an 18-month time period covering all seasons, we particularly look at PsA-EEP impacts at two polar observation stations located at the opposite hemispheres: Tromsø in the NH and Halley in the SH. We find that PsA-EEP can have a measurable impact on ozone concentration above 30 km altitude, with ozone depletion by up to 8% seen in winter periods due to PsA-EEP-driven $NO_x$ enhancement. We also find that direct mesospheric $NO_x$ production by high-energy electrons ($E > 100$ keV) accounts for about half of the PsA-EEP-driven upper stratospheric ozone depletion. A larger PsA-EEP impact is seen the SH where the background dynamical variability is weaker than in the NH. Clearly indicated from our results, consideration of polar vortex dynamics is required to understand PsA-EEP impacts seen at ground observation stations, especially in the NH. We conclude that PsA-EEP has potential to make an important contribution to the total EPP forcing, thus it should be considered in atmospheric and climate simulations.



# 1 Introduction

Atmospheric chemistry impacts of EPP have effectively been studied since 1960s when satellite-based particle flux measure-
ments became available (for a review, see e.g. Sinnhuber et al., 2012). Later evidence on potential climate connection through
middle atmospheric ozone depletion have extended studies to include dynamical coupling processes between the atmospheric
layers (Rozanov et al., 2005; Seppälä et al., 2009; Baumgaertner et al., 2011; Maliniemi et al., 2013). The EPP contribution is
now recognized as an important part of solar forcing in climate simulations (Matthes et al., 2017), particularly when assessing
regional climate variability in the polar regions over decadal time scales.

While the chemical processes leading to middle atmospheric production of ozone depleting catalysts such as odd hydro-
gen ($HO_x$, defined as the sum of H, OH, and $HO_2$ molecules) and odd nitrogen ($NO_x$, defined as the sum of N, NO, $NO_2$
molecules) are well known (e.g. Verronen and Lehmann, 2013), the flux and spectra of EPP that drive these processes remain
partly uncertain. The solar forcing data set prepared for the Couple Model Intercomparison Project Phase 6 (CMIP6) includes
atmospheric ionization rates due precipitation of solar protons, radiation belt electrons ($E = 30 – 1000$ keV), and galactic
cosmic rays (Matthes et al., 2017). The electron ionization data set does not explicitly include the contribution of auroral
electrons ($E < 30$ keV), and it also suffers from shortcomings of the utilized satellite-based observations which lead to larger
uncertainties at electron energies higher than 300 keV and low fluxes (Rodger et al., 2010a; van de Kamp et al., 2016, 2018;
Nesse Tyssøy et al., 2019). In order to understand the extent of these observational uncertainties, there have also been theo-
retical studies providing estimates of local chemical forcing from different types of electron precipitation separately, including
substorm and microburst precipitation (Turunen et al., 2009; Verronen et al., 2013; Seppälä et al., 2015, 2018).

    Polar EPP-$NO_x$ amounts drive the upper stratospheric ozone impact and depend on its production in the mesosphere–lower
thermosphere and transport to lower altitudes (e.g. Sinnhuber et al., 2011). Observations have shown that exceptionally strong
wintertime descent can lead to a 40% – 60% ozone depletion (Randall et al., 1998; Randall et al., 2005). On the other hand, the
relatively large year-to-year variability in atmospheric dynamics makes the overall EPP-$NO_x$ impact less clear especially in the
NH (Päivärinta et al., 2013; Funke et al., 2014). There is also a large variability in year-to-year EPP and its $NO_x$ production,
and also a variability driven by the solar cycle (Andersson et al., 2018). Thus, it is a requirement to capture both the EPP and
dynamics variability in simulations. The challenges emerge especially during disturbed dynamical conditions (Randall et al.,
2015; Funke et al., 2017), and in the representation of electron precipitation forcing (Nesse Tyssøy et al., 2019). Finally, a
detailed description of the lower ionospheric chemistry enhances the $NO_x$ production and the magnitude of the EPP impacts
(Andersson et al., 2016; Kalakoski et al., 2020), but it is not included in most models today.

    Pulsating aurora (PsA) is a type of diffuse aurora which appears in patches of emission or other irregular shapes (Lessard,
2013; Nishimura et al., 2020). While a large number of the emission structures in PsA undergo quasi-periodic intensity mod-
ulations, there are also patches which appear quasi-stable (Grono and Donovan, 2018). Pulsating aurora is most commonly
observed after the magnetic midnight, with a likelihood of about 50% (Jones et al., 2011; Bland et al., 2019). It is also com-
monly related to substorm recovery phases but typically observed to continue beyond the recovery of the magnetic deflection.
PsA is found in the equatorward part of the auroral oval, ranging from about 60° to about 70° magnetic latitude (Grono and



Donovan, 2020; Bland et al., 2020), which magnetically maps to the radiation belt region in the inner magnetosphere. PsA can persist for several hours, up to 15 hours (Jones et al., 2013b), with a median duration of about 2–4 hours. It may also occupy both hemispheres at the same time (e.g. Partamies et al., 2017), as the source region is near-equatorial. These observations

reveal PsA being a very common auroral structure. It does not require particularly strong magnetic activity as a driver, and interestingly, PsA decays slower than the geomagnetic activity recovers. This means that the particle precipitation proxies based on the geomagnetic indices may not capture the long-lived precipitation past the lifetime of the magnetic disturbances.

The particle precipitation during PsA has been reported to cover a large energy range from a few keV up to several hundreds of keV, with the full range modulating in tandem (e.g. Miyoshi et al., 2010, 2015; Grandin et al., 2017). The lower energy end

causes the auroral emission whereas the higher energy end of the observed precipitation penetrates through the mesosphere down to about 60–70 km heights. While this is the bulk behaviour, there is a softer population of PsA precipitation, which does not reach the mesosphere. The dependence of the precipitation energy and the temporal evolution of the PsA patch size (Partamies et al., 2019) as well as the precipitation energy and the morphology of PsA structures (Tesema et al., 2020b) have also been investigated. In the former study, it was concluded that the PsA events with increasing patch sizes do not change

the auroral peak emission height, and would thus not be dominated by energies higher than the usual aurora. The latter study examined PsA events of different morphological sub-categories (amorphous PsA and patchy PsA) and concluded that higher energies are typically observed in association with the quasi-stable patchy PsA as compared to the more transient and largely unstable amorphous PsA.

The high particle energies and the long duration of the PsA precipitation inspired assessments of the mesospheric chemical

changes during PsA. First simulated results of PsA-EEP-related ozone depletion were reported by Turunen et al. (2016). They used a PsA-EEP forcing based on a precipitation spectrum observed by Miyoshi et al. (2015) as an input in the Sodankylä Ion and Neutral Chemistry (SIC) model simulations. As a result of 30 minute precipitation up to 14% of percent of ozone at around 75 km was lost locally during the event due to production of odd hydrogen. Based on ground observations, however, a half an hour duration underestimates the true duration of PsA (Jones et al., 2011; Bland et al., 2019). A more recent study by Tesema

et al. (2020) constructed a median particle precipitation spectrum over the energy range of 30 eV – 1000 keV from about 250 low-altitude spacecraft overpasses during PsA events. These statistical median and extreme spectra were constructed by averaging all electron measurements over the region which was optically defined to be occupied by PsA. Thus, these overpass-average spectra do not take the "patchiness" of the pulsating aurora into account. Using a median duration (120 minutes) of the observed 840 PsA events in the model, the local ozone depletion in the mesosphere reached about 80% for a few hours

during the PsA peak. The SIC model was further run for the low flux scenario, which describes the lower envelope curve of the whole precipitation spectrum bundle. While no observable $HO_x$ and ozone changes were seen in the mesosphere during this experiment, the high flux scenario (upper envelope curve) constructed from the collected spectra resulted in strong depletion (>90%) of mesospheric ozone within a vertically thin layer around 79 km. Thus, these previously reported modelling efforts suggest that the immediate local effect of the PsA-EEP on the atmosphere is significant. The question that remains whether PsA

is common enough to cause an appreciable longer-term effects over a wider range of latitudes and local times, and whether these could be detected by satellite-based observations. Furthermore, an outstanding issue in simulations is the shortcomings in EPP-



related enhancement of wintertime odd nitrogen (e.g. Randall et al., 2015; Pettit et al., 2019). In this context, understanding the PsA-EEP-driven odd nitrogen production could be particularly useful because PsA events are most common in wintertime (Bland et al., 2019). Finally, the PsA-EEP high-energy end can directly increase the mesospheric $NO_x$ production which should
enhance the indirect ozone impact in the upper stratosphere (Sinnhuber et al., 2016; Andersson et al., 2018).

In this paper, we study the chemical impacts of PsA-EEP at polar latitudes. We use the Whole Atmosphere Community Climate Model with its lower ionospheric chemistry extension (WACCM-D), together with a plausible estimate for PsA-EEP forcing based on observations reported in the literature, to simulate the PsA-EEP impact. We analyse the atmospheric response for over an 18-month period, including all seasons of the year. Considering both the Northern and Southern hemispheres,
we select the locations of Tromsø (69.60°N & 19.20°E, 66.64°CGMlat) and Halley (75.58°S & 26.66°W, 65.78°CGMlat) observation stations from our global simulations in order to understand the expected impact that could be measured by ground-based instrumentation such as radiometers (Daae et al., 2014; Newnham et al., 2018), and the next generation EISCAT_3D ionospheric radar system (McCrea et al., 2015). Finally, we discuss our results in the context of overall EPP impact and the current challenges in representing EPP forcing in simulations.

## 2 Model and simulations

Here we use the Whole Atmosphere Community Climate Model (WACCM) version 4, described in detail by Marsh et al. (2007) and Marsh et al. (2013). Running simulations in a specified dynamics mode, the model temperature, horizontal winds, and surface pressure below 50 km were nudged to NASA GEOS5.1 re-analysis data (Rienecker et al., 2008). At the levels above 50 km up to the model upper boundary at $6 \times 10^{-6}$ hPa the model dynamics are free-running, although there is a degree
of control coming from specified dynamics below. We make use of the variant WACCM-D which includes a representation of the lower ionospheric chemistry of both positive and negative cluster ions and was particularly designed for EPP studies (Verronen et al., 2016). WACCM-D captures a full range of observed EPP impacts in the middle atmosphere (Andersson et al., 2016; Kalakoski et al., 2020), in contrast to the standard WACCM which includes only a parameterization of $HO_x$ and $NO_x$ production. Recent D-region studies using WACCM-D include work on seasonal changes in ion composition and comparison
of the latitudinal extent of solar proton events against ionospheric observations (Orsolini et al., 2018; Heino et al., 2019). The background EPP forcing used in our WACCM-D simulations includes solar protons (e.g. Jackman et al., 2008), auroral electrons (Marsh et al., 2007), and galactic cosmic rays (Jackman et al., 2016).

To create a typical PsA-EEP forcing for our simulations, we utilize energy-flux spectra and ionization rates published by Turunen et al. (2016). These are based on ionospheric observations of the EISCAT radar and the KAIRA riometric observations
during a PsA event on the 17th of November, 2012. In their Figure 2, Turunen et al. (2016) presented several different ionization rate profiles which differ especially at altitudes below 80 km due to larger electron flux differences and uncertainties present at high electron energies >100 keV. Selecting the PsA-EEP spectrum that is in good agreement with the statistical median spectrum of Tesema et al. (2020), we make use of the resulting "MCMC median" ionization rate profile. The electron spectrum for that was inverted by Turunen et al. (2016) from the ionospheric data using the Metropolis-Hastings Markov Chain Monte





Carlo (MCMC) method (Haario et al., 2006). The MCMC inversion provided electron fluxes at the energy range of $10 -$
$1000$ keV which leads to atmospheric ionization mainly at altitudes $60 - 125$ km.

In our WACCM-D simulations, the PsA-EEP ionization rates are applied every other night, at magnetic local time (MLT)
hours between midnight and 6 a.m., homogeneously between $60°$ and $72°$ of geomagnetic latitude. This approach neglects
any fine structures in PsA-EEP but still provides our global WACCM-D simulations with realistic and useful forcing scenario.

The applied latitudinal and temporal extent follows the reported, about 50% occurrence frequency of PsA in the morning
sector local times (Bland et al., 2019), although that study did not comment on the PsA occurrence times sequence. The
radar detection study by Bland et al. (2019) further suggested a typical PsA duration of $2 - 4$ hours. However, also extremely
long-lasting events, such as 15 hours by Jones et al. (2013a), have been reported. We therefore settled on a 6-hour duration
for this study, which is on the lengthy side of observed events while still being realistic. The latitude extent used for the

PsA forcing corresponds to the Fennoscandian Lapland latitudes at the equatorward part of the average auroral oval location,
where previous statistical studies have observed a high occurrence rate of pulsating aurora (Partamies et al., 2017; Tesema
et al., 2020). In summary, with these simplifying assumptions we aim at simulating and analysing the full potential of PsA
atmospheric impacts. Note that the same PsA-EEP forcing characteristics are applied throughout the year, which allows for
direct comparisons between seasonal atmospheric responses.

To demonstrate the impact of the PsA-EEP forcing in WACCM-D simulations, Figure 1 shows two snapshots of global
electron concentration on 18th of January at $\approx 78$ km altitude. Overall features include higher values on the dayside ionosphere
from photoionization as well as higher values in the auroral regions due to particle precipitation. Very high electron concentra-
tion are shown with red color and occur at the time and place of PsA-EEP forcing. During every other day, these high-ionization
PsA-EEP patters remain at the same magnetic local times and rotate once around the magnetic poles, following the magnetic

latitudes of the forcing. The locations of Tromsø and Halley stations are marked on the map, both being within the latitude
bands that are directly affected by the PsA-EEP forcing.

We simulate the time period between January 2010 and June 2011 (18 months). Three simulations were made, with different
PsA-EEP forcing scenarios: 1) no-PsA, i.e. zero ionization for a reference; 2) full-PsA, i.e. the MCMC median ionization from
Turunen et al. (2016), 3) thermo-PsA, like full-PsA but the ionization below 85 km ($\approx 4 \times 10^{-3}$ hPa) set to zero. Comparisons

between the full-PsA and no-PsA simulations gives us an estimate of PsA atmospheric impacts, while the thermo-PsA simula-
tion can be used to separate the impacts from thermospheric and mesospheric forcing. All simulations included the background
EPP forcing used in WACCM, i.e. solar protons, auroral electrons, and galactic cosmic rays. Note that the simulation period is
in the ascending phase of the solar cycle right after a record minimum in solar activity, thus the background EPP forcing was
relatively low, making it easier to identify the PsA-EEP impact. For example, maximum daily $Ap$ in the 18-month period was

54.6, as opposed to the maximum $Ap$ of 203.9 for the full cycle of $2001 - 2011$.

The simulations were analyzed for impacts on electron density, $HO_x$, $NO_x$, and $O_3$ concentration in the height range of $10 -$
$120$ km. We especially look at two ground station locations: Tromsø ($69.60°$N & $19.20°$E, $66.64°$CGMlat) and Halley ($75.58°$S
& $26.66°$W, $65.78°$CGMlat). These provide a view on local effects over the opposite hemispheres and in different dynamical
conditions, at two stations that have hosted and will host a wealth of instrumentation for ionospheric and atmospheric research.





Both selected locations are within the latitude band of PsA-EEP forcing, and auroral forcing in general, and thus the results from our 3-D chemistry-dynamics simulations for these locations can be compared to those of Turunen et al. (2016) and Tesema et al. (2020) calculated with a 1-D chemistry model.

## 3 Results

Figure 2 presents the simulation results for the entire 18-month time period at the location of Tromsø in the NH auroral region.
The temporal resolution of the electron, $NO_x$, $HO_x$, and ozone data is one hour, thus the diurnal variations are included. In the panels a – d, the absolute concentrations from the full-PsA simulation display the overall seasonal variability as well as the vertical distributions at 10 – 110 km. The electron concentration increases towards higher altitudes due to the increasing ionization from the solar short-wave radiation and EPP. $NO_x$ displays two characteristic maxima, one in the stratosphere (at $\sim 30$ km) and another in the lower thermosphere (at $\sim 110$ km), and more (less) $NO_x$ in the mesosphere (stratosphere) during
wintertime. Denitrification due to reactions with chlorine leads to very low concentrations in the lower stratosphere around 20 km during winter periods. The $HO_x$ concentration is higher during summer due to its production being driven by solar ultraviolet and Lyman-$\alpha$ radiation, and maximizes around the stratopause (at $\approx 50$ km). In wintertime, largest concentrations are seen in the mesosphere. Ozone has two maxima in the summer: the stratospheric ozone layer peaking at 20–30 km and the secondary maximum at mesopause at around 90 km. In wintertime, the mesospheric ozone concentration is higher than in
the summer due to less loss from diminished solar radiation and photodissociation. The tertiary maximum development in the middle mesosphere around 70 km at the vicinity of the polar winter terminator contributes to higher wintertime concentrations as well.

Highlighting the PsA-EEP impact, differences between the full-PsA and no-PsA simulations are shown in the panels e – h of Figure 2. The electron concentration clearly enhances at 60 – 90 km during summer/daytime due to the added PsA-EEP
ionization. In general, in the wintertime and at night there is much less impact than in the summer. However, because of the absence of solar radiation, a larger portion of the negative charge below 90 km is held by ions, not electrons (e.g. Verronen et al., 2016; Orsolini et al., 2018). Above 90 km, the wintertime differences exhibit a variation between increases and decreases which are relatively small and are related to the internal variability of the model coming from its free-running dynamics at the upper altitudes. Also, the ionization from auroral electrons, which is the same in all simulations, becomes dominant over
PsA-EEP at altitudes above $\approx 90$ km (not shown). The $NO_x$ concentrations show a similar, relatively small variability around 100 km, but only during the winter periods when there is more dynamical variability. Focusing on the main $NO_x$ features, increases are seen at 80 – 90 km throughout the year. Early in the winter season, the $NO_x$ increase due to PsA-EEP is observed in the mesosphere, from where it further descends into the stratosphere reaching down to about 20 km by the end of the winter season. The increase related to the descent disappears and then appears several times during the winter. A layer of PsA-EEP
$NO_x$ persists at about 25 km altitude until the end of the simulation period (midsummer). The $HO_x$ response shows some diurnal variability, i.e. cyclic increases and decreases. Overall, however, there is an increase of $HO_x$ at 70 – 80 km altitudes from the direct PsA-EEP impact. Around 60 km, the $NO_x$ increase leads to chemical loss of wintertime $HO_x$ concentrations





(see e.g. Verronen and Lehmann, 2015). The major feature in the ozone response is the descending stratospheric depletion during the winter and spring seasons which closely follows the descent of enhanced $NO_x$ and is caused by the $NO_x$-driven

ozone loss reactions. Note that the relative ozone response is larger in the mesosphere than in the stratosphere because of lower background concentrations (not shown).

The panels i – l of Figure 2 show comparisons similar to the panels e – h but between the thermo-PsA and the no-PsA simulations. This comparison allows us to assess the importance of the PsA-EEP thermospheric contribution separately. Clearly, the removal the ionization at the bottom part of the ionospheric column removes nearly all of the electron density response from

those altitudes. Thus the ionospheric response is largely restricted to the altitudes which experience direct PsA-EEP forcing. However, $HO_x$, $NO_x$, and ozone show a clear response also at altitudes below 85 km. For all three, the main features of the PsA-EEP impact remain. Particularly, the $NO_x$ descent and the corresponding stratospheric ozone depletion are still clear, even without direct PsA-EEP forcing in the mesosphere. The notable difference is in the magnitude of the response which is smaller than with the mesospheric PsA forcing. Also, the maximum response is reached later in the winter/spring because of the lack

of early winter impact from direct mesospheric $NO_x$ production

Figure 3 displays a similar set of simulation results as in Figure 2, but for the Halley station location in the SH. Based on previous studies (e.g. Andersson et al., 2018), the same EEP forcing is expected to produce a larger wintertime effect on the SH middle atmospheric chemistry, because atmospheric wave-driven dynamical variability is much weaker in the SH than in the NH, which allows for stronger signatures of the chemical response. Indeed, the $NO_x$ response is clearly stronger and displays

much less disruptions during the descent than that seen in the NH. Similar differences between the NH and the SH are are seen in the ozone response as well. Additional differences compared to the NH response are the descending depletion of $HO_x$ at 20 – 40 km and the ozone increase at 15 – 30 km, both seen from mid winter to spring. The former is caused by conversion of $HO_x$ to $NO_y$ species, such as $HNO_3$, in reactions with enhanced $NO_x$. The latter is due to enhanced denitrification which converts ozone-depleting, active chlorine and bromine to reservoir species. As in the NH, the removal of the PsA-EEP direct

mesospheric impact leads to a decrease in the magnitude of the atmospheric response below 85 km and about a one-month delay in the maximum stratospheric ozone response.

Figures 4, 5, and 6 show the height-integrated PsA-EEP impact on $HO_x$, $NO_x$, and ozone columns at altitudes >30 km and >60 km. Overall, the hourly responses display a large variability in the mesosphere, such that the impact there is somewhat masked. For $NO_x$ and $O_3$, the inclusion of upper stratosphere makes the impact much clearer. To clarify the PsA-EEP impact in

cases with large variability, we have calculated 30-day running averages from the hourly data and show also them in Figures 4, 5, and 6.

Without the direct mesospheric forcing, the PsA-EEP impact on $HO_x$ does not reach 10% and is clearly negligible compared to the overall variability (Figure 4). When the mesospheric forcing is included, there is a clear increase in $HO_x$ during the winter periods, with the 30-day impact peaking at about 30% in the SH mesosphere. The inclusion of the upper stratosphere reduces

the maximum 30-day impact to about 25% in the SH and to about 15% in the NH. Note that in the hourly data the impact can temporarily reach beyond 100% in both hemispheres and altitude ranges.



The PsA-EEP forcing leads to a mesospheric 30-day mean $NO_x$ increase of about 20% during the summer periods (Figure 5). In the winter periods the increase is larger, reaching about 100% in the SH and about 50% in the NH. Note that in the NH the response is clearly larger in the second winter than in the first one, mainly because the simulation and the PsA-EEP forcing begin in January and only cover the latter half of the first winter. The exclusion of the direct mesospheric PsA-EEP forcing reduces the impact especially in the winter periods, by up to a factor of two to three. The inclusion of the upper stratosphere reduces the variability in the hourly data, so that the impact becomes clear: in wintertime $NO_x$ increases while during summer periods the PsA-EEP impact is negligible. The wintertime peak increase is 50% – 75% in the SH, reaching beyond 100% temporarily. In th NH, the peak increase during the second winter is around 50% while showing a larger dynamical variability over the winter period. Again, the exclusion of the direct mesospheric PsA-EEP forcing leads to a smaller maximum impact, i.e. around 40% in the SH and 30% in the NH.

The mesospheric (>60 km) 30-day mean ozone response is small in the NH, a decrease of a few percent is seen in the winter periods (Figure 6). In the SH, the ozone decrease reaches up to about 6% in mid-winter. The exclusion of mesospheric PsA-EEP forcing reduces the ozone decrease to less than 1% for all seasons, while there is an increase of a few percent in the winter periods. The lack of a clear ozone decrease without the mesospheric PsA-EEP forcing is in line with the negligible $HO_x$ response because $HO_x$ is the main ozone loss catalyst in the mesosphere. As seen in the hourly data, the mesospheric ozone response shows a large overall variability ($> \pm 10\%$), even in the SH with the direct mesospheric PsA-EEP forcing included.

When including the upper stratosphere (>30 km), the ozone response becomes clear in the hourly data (Figure 6). In the full-PsA forcing scenario, a NH column ozone depletion of up to 3% is seen around the middle of the second winter (black curve). This response is related to the PsA-$NO_x$ descent. No real ozone response is seen during the first winter because simulation begins in January, i.e. in the middle of that winter, leaving less time for PsA-$NO_x$ production and descent. If the mesospheric PsA-EEP forcing is excluded (cyan curve), the wintertime peak ozone depletion reduces to around 1%, i.e. the impact is reduced by a more than a factor of two. In the SH the impact is qualitatively similar to the NH but the ozone decrease is more consistent over time and also stronger, i.e. 4% – 8% at the end of September. When the mesospheric PsA-EEP forcing is excluded, the PsA-EEP impact reduces to 2% – 3%. Following a full winter, an 0.5% decrease persists over the summer period. Compared to the NH response with the full-PsA forcing, the SH ozone depletion reaches a similar magnitude with thermo-PsA only because of less variability in the polar vortex dynamics in the SH leads to less interruptions in the $NO_x$ descent. The interruptions in the NH are clear, also from the $NO_x$ and $O_3$ data shown in Figure 2. In the full-PsA simulation, the SH ozone depletion is more than double the NH depletion, and both last over the winter season. In both hemispheres, it is clear that the mesospheric $NO_x$ production plays a key role in depleting the ozone, and is important for both the magnitude and the timing of the impact.

In the above analysis we are focusing on the local PsA-EEP impact at the selected stations of Tromsø and Halley. To put our results into a wider context, we next consider the overall polar upper stratospheric impact of PsA-EEP. We do this for the NH only, because there the dynamical variability of the polar vortex is stronger and more drastic. A particularly interesting period in our simulations is January, 2011, when the $NO_x$ and $O_3$ responses disappear and reappear during the month (see Figures 2, 5, and 6).



Figure 7 presents the relative $NO_x$ and $O_3$ response at $\approx$ 40 km altitude, i.e. in the upper stratosphere. At the beginning of January, Tromsø is located within the polar vortex. There, the PsA-EEP impact is clearly seen as increased $NO_x$ and decreased $O_3$ concentrations. There is considerable variability of the relative impact within the vortex with a range of responses up to
about $+180\%$ and down to about $-6\%$, respectively. In the middle of January, the polar vortex has moved away from Tromsø. Although the PsA-EEP impact inside the vortex is quite similar compared to the situation in the beginning of the month, this time none of it can be seen at the Tromsø location. The situation changes back at the end of January when the polar vortex is over Tromsø again. Thus the variability in the $NO_x$ and $O_3$ response seen in Figures 2, 5, and 6 is due to the evolution of the NH polar vortex over the winter period. Clearly, a global model like WACCM is a powerful tool when interpreting results
from a single polar station like Tromsø, as demonstrated here. Although not shown, the SH vortex is much more stationary with respect to the Halley location and the PsA-EEP impacts there do not display similar large variability.

## 4   Discussion

The particle forcing used in this study was recently validated by a statistical analysis of in-situ particle spectra from low-altitude spacecraft measurements (Tesema et al., 2020). It was concluded that the spectrum does indeed represent the observed
median spectrum for PsA particle precipitation very well. The simulation results presented in this study thus provide insight into the effects of the median PsA-EEP forcing. Tesema et al. (2020) also showed that while the low flux PsA forcing causes no atmospheric changes, the high flux PsA forcing could severely deplete mesospheric ozone. It is therefore desirable to investigate both the atmospheric sensitivity threshold towards the low flux scenario as well as the $NO_x$ production and descent during the high flux forcing. As reported by Partamies et al. (e.g. 2017), the solar wind driving during pulsating aurora does
not need to be extreme although the wind speed is typically elevated. This can be particularly important for simulation runs during the solar minimum and the declining phase of the solar activity, because these time periods are known to associate with a frequent high-speed streams in the solar wind (Asikainen and Ruopsa, 2016). The question is whether PsA forcing during consecutive nights would lead to a stronger cumulative effect in the atmosphere than what we have seen in this study. Some variations in the PsA MLT extent is expected due to changes in the solar wind driving. The latitude extent of PsA, which maps
to the ionosphere from the outer radiation belt source region, is likely to undergo little variability from event to event (Sandhu et al., 2019). However, the outer radiation belt electron flux at high L-shells has been observed to increase during the solar declining phase and the minimum phase (Miyoshi et al., 2004), which will affect the latitude extent of a longer-term PsA-EEP forcing. Thus, a more thorough analysis of the spatial and MLT extent should be done in order to estimate how realistic is the latitude and MLT extent of the PsA-EEP forcing used in this study. Furthermore, our results suggest that the overlap between
the polar vortex and the particle precipitation region is a key factor in determining the chemical impact. It is therefore important not to study the impact area in isolation but with respect to the vortex location and area.

Based on our analysis, PsA-EEP is common and strong enough to have an impact on polar ozone through the production of $NO_x$ in the mesosphere–lower thermosphere and its wintertime descent inside the polar vortex. The magnitude of the mesospheric column response from the $HO_x$ increase, i.e. up to about 5% decrease above 60 km in the SH (Figure 6d, 30-





day mean), is clearly smaller than the PsA-EEP impact estimated in previous studies (Turunen et al., 2016; Tesema et al., 2020), and smaller than what has been estimated for substorm precipitation (Seppälä et al., 2015). However, these previous studies considered the direct short-term ozone response in the middle mesosphere locally, and did not consider the background variability from atmospheric dynamics in their simulations. Our results now indicate that satellite-based detection of PsA-EEP ozone impact would be challenging in the mesosphere due to overall short-term variability of the response. Even in simulations,

the variability of internal model dynamics could largely mask ozone responses smaller than $\approx 5\%$ (e.g. Verronen et al., 2020). In the column above 30 km, including the upper stratosphere, the ozone decrease reaches up to 8% and 3% with and without the direct mesospheric PsA-EEP impact, respectively. This highlights the importance of the mesospheric $NO_x$ production to the stratospheric ozone response, and is in agreement with previous studies (Arsenovic et al., 2016; Andersson et al., 2018). For example, Andersson et al. (2018) noted a similar ozone response to mesospheric $NO_x$ production when applying the CMIP6

MEE forcing. It is also interesting to note that the magnitude of our PsA-EEP ozone response is close to the 7% ozone decrease from MEE reported by Andersson et al. (2018), although our model setup and analysis is different from theirs. Particularly, Andersson et al. (2018) made their simulations with fully free-running dynamics, made their comparisons between years of high and low EPP, and did not use a D-region ion chemistry extension. All of these differences should contribute to smaller atmospheric responses than in our analysis. Since the CMIP6 MEE forcing is from a proxy model based on observations,

it should cover all types of electron precipitation at energies 30 – 1000 keV including PsA-EEP. Thus, the similarity in the magnitude of ozone response could indicate an overestimation of PsA-EEP impact, or an underestimation of overall electron impact in the CMIP6 MEE set as suggested e.g. by Nesse Tyssøy et al. (2019). On the other hand, a larger EPP impact is generally seen in satellite observations, e.g. Fytterer et al. (2015) and Damiani et al. (2016) have reported ozone depletion between 5% and 15% in the upper stratosphere. Therefore, in the context of the current underestimation of EPP-$NO_x$ in

simulations (Hendrickx et al., 2018; Smith-Johnsen et al., 2018; Pettit et al., 2019), PsA-EEP could provide part of the missing $NO_x$ and improve simulations of ozone response.

Although we present a simplified sensitivity study here, the results indicate that PsA-EEP has the potential to contribute considerably to the total EPP forcing and a stronger response in middle atmospheric $NO_x$ and ozone. Currently, the CMIP6 MEE forcing data cover part of our estimated PsA-EEP energy and latitude range (van de Kamp et al., 2016). However, there

are problems related to the satellite-based data used to create CMIP6 MEE, including proton contamination and noise floor issues. Also, the spatio-temporal variability of CMIP6 MEE is driven by the geomagnetic $Ap$ index. It is not clear how good a proxy $Ap$ is for different types of EPP because PsA, substorm precipitation, and microburst electrons all have their own energy, latitude, and temporal characteristics and are driven by different solar wind and magnetospheric processes (Asikainen and Ruopsa, 2016). Understanding quantitatively the relative contributions of different types of EPP to the total forcing remains

a challenge.

Finally, we note that our study presents the atmospheric response to a simplified, repeating pattern of PsA-EEP forcing and highlights some seasonal differences between the NH and the SH. In each hemisphere, however, seasonal responses are expected to vary from year to year, driven by variations in PsA-EEP forcing and modulated by differences in polar atmospheric dynamics which define the background conditions locally (e.g. Newnham et al., 2018). Over longer periods, assessment of the

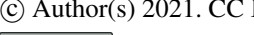



PsA-EEP impact on the atmosphere and further on climate requires understanding of the PsA-EEP variability over solar cycle time scales and consideration of the year-to-year variability of dynamical conditions. This is a target for future studies in which chemistry-climate models like WACCM-D are a strong asset.

## 5 Conclusions

In this paper, we have studied the seasonal ozone impacts of PsA-EEP, particularly at two polar observatory locations. Based on
our simulations which utilize latest knowledge on PsA-EEP energy-flux spectrum and spatio-temporal extent, and the WACCM model with its lower ionospheric chemistry extension, we conclude the following:

1. PsA-EEP has the potential to cause a measurable ozone depletion in the column above 30 km. The main impact, up to 8% decrease in the upper stratosphere in winter periods, is caused by descent of PsA-$NO_x$ from altitudes above. The mesospheric production of $NO_x$ from the high-energy part of PsA-EEP causes more than half of the ozone response.

2. In the mesosphere, there is a PsA-EEP 30-day mean impact reaching to 5% of $O_3$ ozone loss from the $HO_x$ enhancement. However, on hourly time scales the ozone response is less clear and displays a large variability.

3. A larger PsA-EEP impact is seen in the SH where the variability of atmospheric dynamics is smaller than in the NH. Overall, the interpretation of ground-based observations requires consideration of polar vortex dynamics.

We conclude that PsA-EEP has the potential to contribute to the total EPP forcing and lead to a stronger response of middle
atmospheric $NO_x$ and ozone. More work is needed to understand qualitatively the relative contributions from different types of EPP to the atmospheric response over solar cycle time scales.

*Code and data availability.* WACCM source code is distributed freely through a public subversion code repository of the Coupled Earth System Model CESM (http://www.cesm.ucar.edu/models/cesm1.0/, UCAR, last access in April 2021). WACCM-D has been officially released with the CESM version 2.0 in June 2018 (http://www.cesm.ucar.edu/models/cesm2/, UCAR, last access in April 2021). The
simulation data used in the analysis are open-access and freely available through the B2SHARE service (http://doi.org/10.23728/fmi-b2share.d3bfeca00c1744328da71cc7a8dcad76, FMI, last access in May 2021).

*Author contributions.* All authors contributed to the original research plan. PTV, AK, and NP defined the PsA forcing. PTV and MES made the simulations. PTV and AK analysed the data and made the figures. PTV, AK, and NP led the writing of the paper. All authors contributed to the final paper.

*Competing interests.* Y. Miyoshi is a topical editor of ANGEO.





*Acknowledgements.* The work of P. T. Verronen and M. E. Szeląg is supported by the Academy of Finland (project No. 335555 ICT-SUNVAC). The work by N. Partamies is partly supported by the Norwegian Research Council (NRC) under CoE contract 223252 and NRC contract number 287427. The work of A. Kero is funded by the Tenure Track Project in Radio Science at Sodankylä Geophysical Observatory/University of Oulu.





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



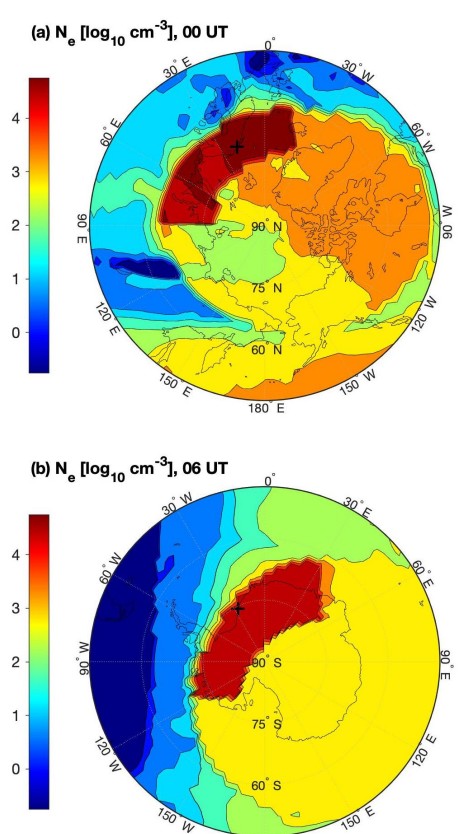

**Figure 1.** An example of simulated electron concentrations on 18th of January, 2010, at ≈78 km altitude. The locations of the Tromsø and Halley stations are marked with the black crosses in the NH (a) and the SH (b), respectively.



**Figure 2.** Simulation results for a selected location in the Northern Hemisphere, namely the Tromsø radar site (69.6°N, 19.2°E). *Leftmost column*: (a) electron concentration $N_e$, (b) $NO_x$, (c) $HO_x$, and (d) $O_3$ from the full-PsA simulation. *Center column*: Absolute differences in (e) $N_e$, (f) $NO_x$, (g) $HO_x$, and (h) $O_3$ between the full-PsA and no-PsA simulation. *Rightmost column*, panels (i) to (l): as the center column, but showing differences between the thermo-PsA and no-PsA simulation.







**Figure 3.** Model results for a selected location in the Southern Hemisphere; Halley Station in Antarctica (75.6°S, 26.6°W). For a description of the panels, see the caption of Figure 2.

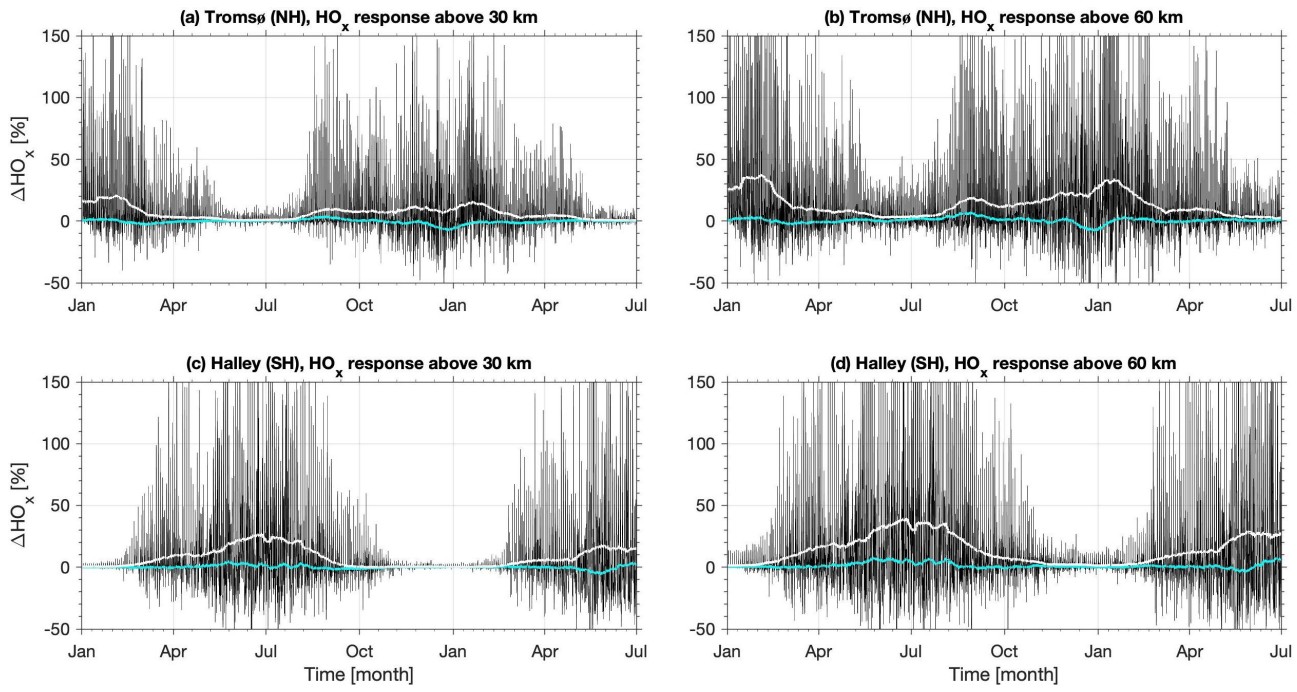

**Figure 4.** Height-integrated $HO_x$ response to PsA-EEP at the Tromsø and Halley stations. (black line) The difference between the full-PsA and no-PsA simulations in one-hour resolution. (white line) The 30-day running mean of the black line. (cyan line) The 30-day running mean of the difference between thermo-PsA and no-PsA simulations.



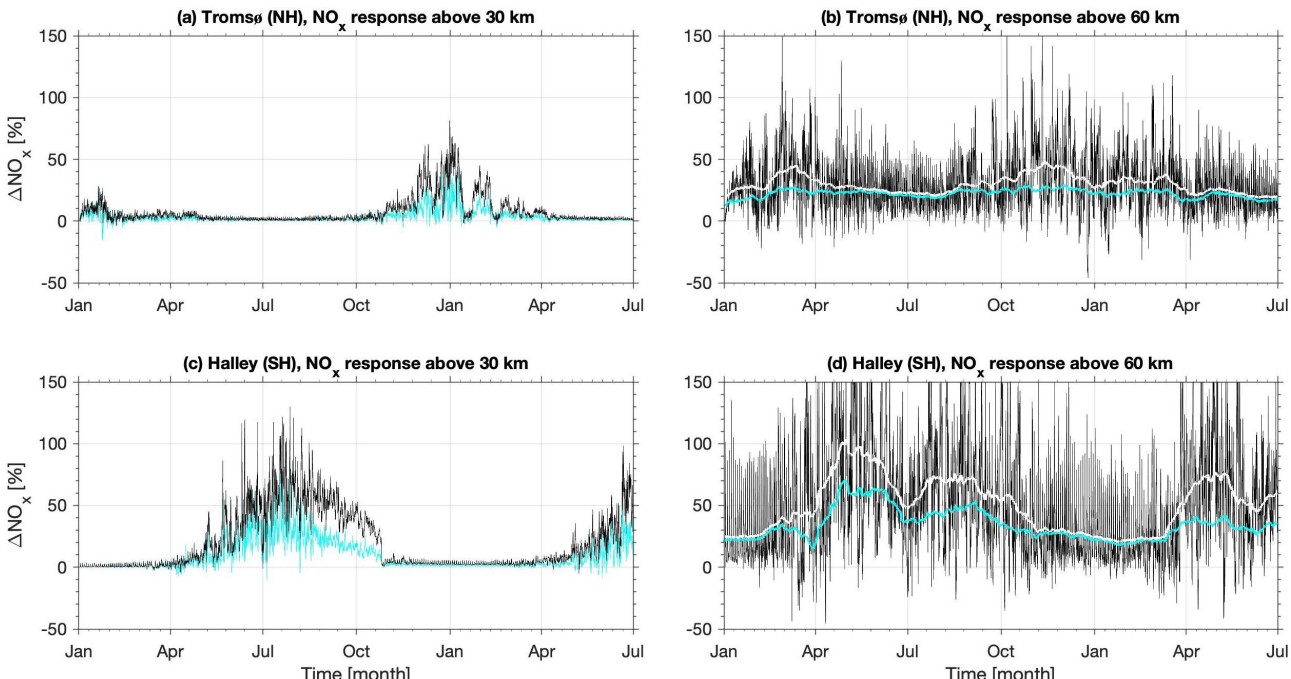

**Figure 5.** Height-integrated $NO_x$ response to PsA-EEP at the Tromsø and Halley stations. See Figure 4 caption for explanations.

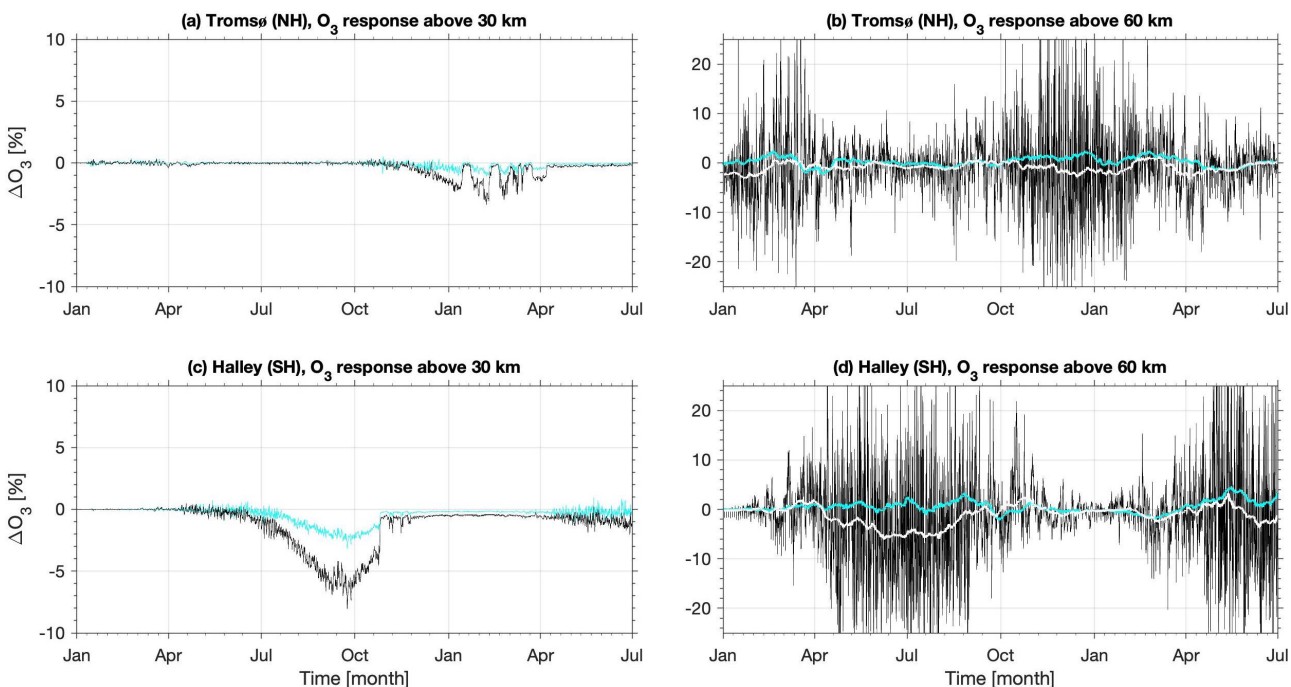

**Figure 6.** Height-integrated $O_3$ response to PsA-EEP at the Tromsø and Halley stations. See Figure 4 caption for explanations.



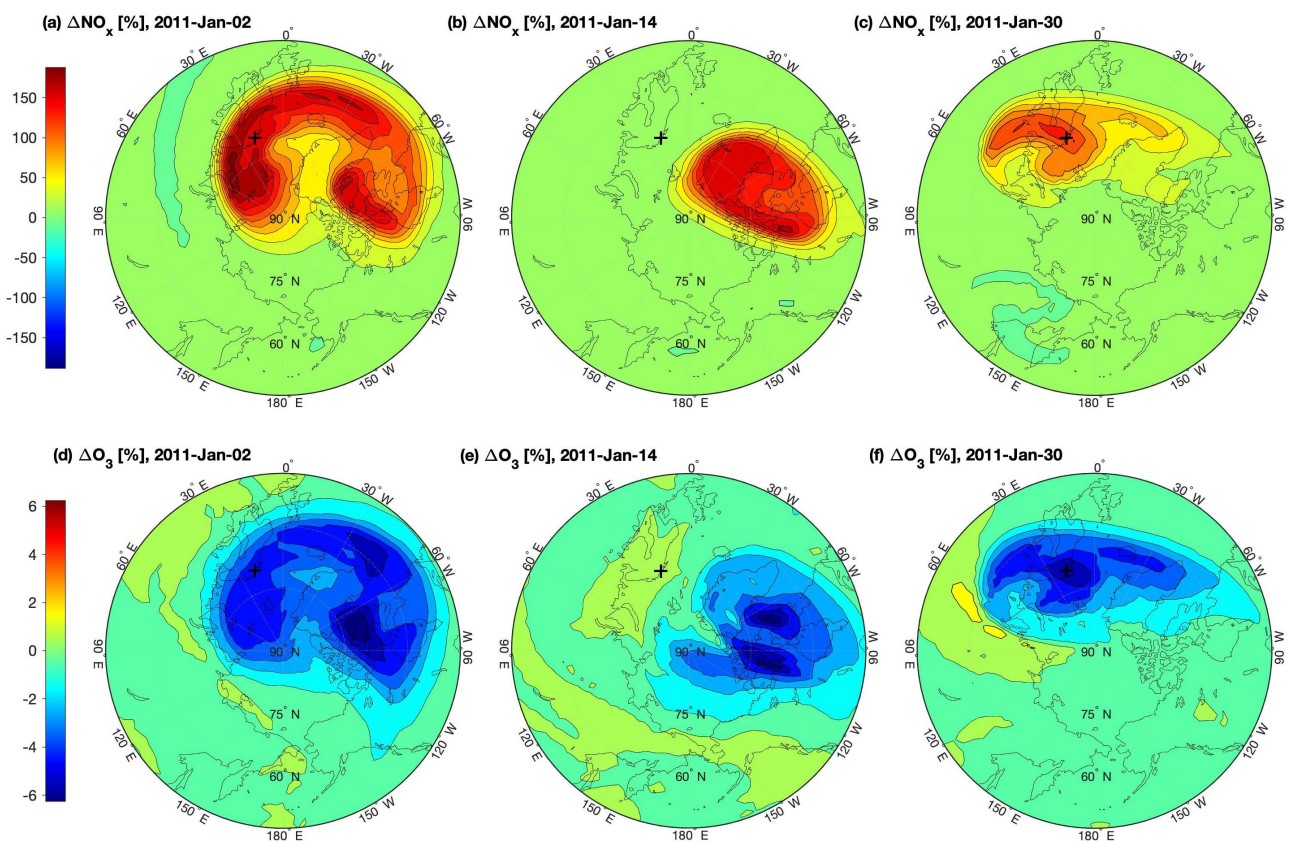

**Figure 7.** (top row) Relative $NO_x$ differences between the full-PsA and no-PsA simulations at ≈40 km altitude, at 12 UT on (a) 2011-Jan-02, (b) 2011-Jan-14, and (c) 2011-Jan-30. The black cross indicates the Tromsø station location. Latitudes from $45°N$ to $90°N$ are displayed. (bottom row) As the top row, but for $O_3$.