# Peer review of "Simulated seasonal impact on middle atmospheric ozone from high-energy electron precipitation related to pulsating aurorae"

_Annales Geophysicae, 2021_

## Author Comment (AC1)

**Authors' response to reviewers' comments on "Simulated seasonal impact on middle atmospheric ozone from high-energy electron precipitation related to pulsating aurorae" by Verronen et al.**

Please find below our answers (in blue) to the comments (in black).

**Response to the comments of Referee #1**

General Comments: The article "Simulated seasonal impact on middle atmospheric ozone from high-energy electron 5 precipitation related to pulsating aurorae" by Verronen et al. presents the results of model simulation on the impacts of the energetic electron precipitation related to pulsating aurorae (PsA-EEP) to the middle atmospheric ozone. However, there are several issues need to be addressed before the consideration of the publication.

Response to general comments: We would like to thank the reviewer for the comments and appreciate the time devoted to the 10 evaluation of our paper.

1. Regarding "There is also an open question of the relative contribution of PsA-related energetic electron precipitation (PsA-EEP) to the total atmospheric forcing by solar energetic particle precipitation (EPP)" in the abstract, why did you choose the pulsating aurora over the ordinary diffuse or discrete aurorae to investigate the EPP impact on ozone chemistry in the polar mesosphere? What makes the pulsating aurora special over the ordinary aurorae, in particular, with regard to their atmospheric impacts? How are PsAs distinguished from the ordinary diffuse aurora or discrete aurora in terms of energy,

occurrence rate and duration, latitudinal extent etc. to affect the atmospheric chemistry and possibly dynamics? It is critical to justify the results of the study especially when the simulation impacts of PsA-EEP seem to be similar to typical EPP impacts.

Response: Electron precipitation related to pulsating aurorae (PsA-EEP) has recently received increasing research attention because of its different characteristics compared to other types of aurorae. Specifically, PsA-EEP extends to higher electron

- 20 energies than any other morphological type of auroral precipitation and thus leads to direct ozone depletion in the mesosphere through  $HO_x$  and mesospheric production of  $NO_x$  (Turunen et al., 2016, Miyoshi et al., Penetration of MeV electrons into the mesosphere accompanying pulsating aurorae, Sci. Rep., 11, 13724, https://doi.org/10.1038/s41598-021-92611-3, 2021). Further, PsA-EEP has a higher occurrence rate and longer duration (Bland et al., 2019), and greater latitudinal extent than other auroral precipitation (Bland et al., 2021). All these PsA-EEP features contribute towards stronger atmospheric impacts,
- and justify our focus on PsA-EEP here. 25

15

In the revised manuscript (Section 1), we discuss these special PsA characteristics and atmospheric implications.

2. How was the PsA-EEP taken into account within the model as an energy inputs during the simulation period of 18 months in comparison with other EPP energy inputs? Please describe it specifically. In Section 2, the PsA-EEP ionization rates applied in the study were described such as its application of every other night at the local time hours of 00 MLT to 06 MLT, 30 homogeneously between 60 and 72 deg geomagnetic latitude. I am wondering how this application of PsA-EEP forcing is realistic and how different it is from regular diffuse and discrete auroral forcing. Authors mentioned that all simulations included the background EPP forcing used in WACCM, i.e. solar protons, auroral electrons, and galactic cosmic rays. How the applied PsA-EEP forcing is distinguished from these background EPP forcing? This seems to be critical to explain the difference between the full-PsA and no-PsA simulations in Figure 2, assuming that the no-PsA simulation includes all the

background EPP forcing. 35

> Response: We argue that the characteristics of our PsA-EEP forcing are realistic. The occurrence rate, latitude extent, and MLT distributions are based on most recent research on PsA and well justified. Nevertheless, simplifications have been made to create a usable forcing data set for the simulations. Particularly, the well-known rapid pulsating patterns (patchiness) are

not taken into account, as their implementation in the model would be challenging without detailed knowledge of their

- 40 statistical characteristics. Determination of these characteristics would require a separate, extensive effort (e.g., an artificial intelligence application on large amounts of PsA data). However, the simplifications we made are justified by the aim of assessing the potential impact of PsA-related EEP toward its upper limit. It should be noted that similar simplifications of latitudinal extent and temporal variability are typical in simulations for other types of EEP as well.
- The background auroral forcing used in our simulations is provided by a statistical model driven by the geomagnetic Kp
  index. The Kp-aurora is described by, e.g., Smith-Johnsen et al., JGR Space, 123, 5232–5245, doi:10.1029/2018JA025418, 2018, and references therein. In the current setup, it provides daily average ionization rates around geomagnetic latitudes 65°–70° (auroral oval). The Kp-aurora has a characteristic energy of 2 keV and a Maxwellian distribution and it is restricted to altitudes above 90 km (*E* < 30 keV), i.e. it has no direct ionization impact in the mesosphere. With the applied PsA-EEP energy range of 10–1000 keV, PsA-EEP provides a major extension of EEP into mesospheric altitudes. Compared to the</li>
  Kp-aurora, there is also an increase in latitudinal extent by a few degrees, and a stronger diurnal variability from the MLT
- 50 Kp-aurora, there is also an increase in latitudinal extent by a few degrees, and a stronger diurnal variability fro dependency.

The background GCR ionization rates are the daily average values recommended for the Coupled Model Intercomparison Project (Matthes et al. 2017). GCR affects all latitudes and is stronger in the polar regions. However, the GCR ionization peaks in the lower stratosphere and day-to-day variability is small. Compared to PsA, it provides very low, nearly-static background ionization in the mesosphere and above.

The background solar proton ionization rates are the daily average values recommended for the CMIP6 project (Matthes et al. 2017). In our simulations, proton ionization affects latitudes above 60°, geomagnetic, and altitudes below about 90 km. Proton background is small at most times, but can completely dominate the mesospheric and upper stratospheric ionization during major events. During our simulation period, however, there were no large solar proton events and thus proton ionization from the PsA EEP ionization (ftp://ftp.gupe.pose.gov/pub/indices/SPE txt\_NOAA\_accessed

60 ionization remained low compared to the PsA-EEP ionization (ftp://ftp.swpc.noaa.gov/pub/indices/SPE.txt, NOAA, accessed in May, 2021). We note that it is important for our study to include the background GCR, SPE, and aurora, so that PsA-EEP additional impact can be assessed realistically. For this purpose, the background EEP that we have applied is well-suited.

In the revised manuscript, Section 2, we have added a paragraph that discusses the background EPP input and how the PsA-EEP adds to it, in order to clarify the difference between the full-PsA and no-PsA simulations.

- 65 3. At the end of Section 1, authors raised a few questions as "The question that remains whether PsA is common enough to cause an appreciable longer-term effects over a wider range of latitudes and local times, and whether these could be detected by satellite-based observations. Furthermore, an outstanding issue in simulations is the shortcomings in EPP-related enhancement of wintertime odd nitrogen. In this context, understanding the PsA-EEP-driven odd nitrogen production could be particularly useful because PsA events are most common in wintertime. Finally, the PsA-EEP high-energy end can directly
- 70 increase the mesospheric  $NO_x$  production which should enhance the indirect ozone impact in the upper stratosphere." Are these questions answered by this study? What are the fundamental differences of the current study from previous studies?

Response: Yes, we show that PsA-EEP can potentially have appreciable longer-term atmospheric impacts, and that part of the  $NO_x$ -shortage found in previous simulation work could be covered by considering PsA-EEP. The novelty to the previous PsA-EEP studies is that we consider long-term impacts using a global chemistry-climate model, thus the impact of atmospheric dynamics on the  $NO_x$ /ozone response is included e.g. through the  $NO_x$  transport.

We have revised the end of Section 1 and Conclusions to clarify these points.

55

75

4. Regarding thermo-PsA simulation, it was mentioned to separate the impacts from thermospheric and mesospheric forcing. However, it seems very unrealistic and artificial to me. Can it be regarded as the pulsating aurora with relatively low-energy electrons? If so, why didn't you set two different PsA EPPs with high and low energy, instead of artificially setting the

80 zero-ionization below 85 km? This way should be more physically consistent within the model.

Response: In the thermo-PsA simulation, we set the PsA-EEP ionization below 85 km to zero. A similar forcing scenario could be achieved by setting the electron flux to zero at energies larger than about 40 keV. However, that approach would also remove the along-path thermospheric ionization due to >40 keV electrons. Because our aim is to separate the response to direct thermospheric and mesospheric forcing, our approach does this without losing the high-energy electron impact in the thermosphere (i.e. the thermospheric ionization in full-PsA and no-PsA simulations is the same). While our approach is somewhat artificial, it, however, provides us a useful way to assess the importance of direct mesospheric ionization.

85

In the revised manuscript (Section 2), we discuss this and justify our approach better.

5. Regarding the solar activity effects in the study, authors mentioned as "Note that the simulation period is in the ascending phase of the solar cycle right after a record minimum in solar activity, thus the background EPP forcing was relatively low, making it assign to identify the PoA. EEP impact" Does it mean that the occurrence of PoA is not affected by solar activity.

90

95

making it easier to identify the PsA-EEP impact." Does it mean that the occurrence of PsA is not affected by solar activity while the background EPP forcing is weak during low solar activity? Is PsA different from background EPP in terms of solar activity dependence?

Response: We mean that the selected time period had a low solar activity, so that the background EPP, e.g. Kp-Aurora, was relatively low compared to what they would be closer to solar maximum. Thus if the same PsA-EEP forcing was applied during solar maximum, we would expect to have relatively smaller response compared to the background EPP. However, the selection of a solar minimum time is inline with our approach of finding the upper range of the PsA-related impact. While PsA does not require strong or extreme solar wind driving (Tesema et al., 2020), which means that PsA reduces less than other aurorae during solar minima, it still varies with solar activity similar to other types of aurora.

In the revised manuscript, we clarify this issue in Section 2.

---

## Author Comment (AC2)

**Authors' response to reviewers' comments on "Simulated seasonal impact on middle atmospheric ozone from high-energy electron precipitation related to pulsating aurorae" by Verronen et al.**

Please find below our answers (in blue) to the comments (in black).

**Response to the comments of Referee #2 (Katharine Duderstadt)**

5    General Comments: This study uses observations of pulsating aurora location, occurrence rates, and spectra to estimate potential impacts to atmospheric $HO_x$, $NO_x$, and O3. This perspective is especially important in understanding the 'indirect effect' of mesospheric $NO_x$ enhancements and descent on reductions of O3 in the upper stratosphere. The authors provide a useful comparison between WACCM studies using PsA-EEP and CMIP6 MEE on atmospheric ionization and composition. The paper is well-written and figures are clear.

10    Response to the general comments: We thank the reviewer for her comments. We also appreciate the time devoted to the evaluation of our paper.

Specific Comments:
1. It would be valuable to place PsA-EEP estimates in context of what is known (and not known) about radiation belt electron precipitation. Since the PsA-EEP driven WACCM results are so close to the CMIP6 MEE simulations, does this imply that
15   most of the electron precipitation from the outer radiation belt should produce pulsating aurora? Are there other mechanisms for precipitation that do not result in PsA but are observed by polar orbiting satellites? How are PsA-EEP related to substorm and microburst precipitation? How might the results of this study inform our understanding of electron precipitation processes from the radiation belts?

Response: PsA-EEP is related to other types of electron precipitation and cannot be fully separated in satellite-based electron
20   flux observations. According to the current understanding, the primary cause of PsA is electron precipitation from the plasma sheet and the inner magnetosphere (outer radiation belt) (Nishimura et al., Space Science Reviews, 216, 4, 2020). PsA-EEP is often observed during substorm activity but also extends beyond substorm disturbance and includes higher-energy electrons than typical substorm precipitation. A relation between PsA and microburst precipitation is expected theoretically (Miyoshi et al. Geophys. Res. Lett., 47, e90360, doi:10.1029/2020GL090360, 2020), but has not been observed and is not understood in
25   detail (Miyoshi et al., J. Geophys. Res., 120, 7728–7736, doi:10.1002/2015JA021562, 2015b).

In the revised manuscript, we discuss this in Section 1.

Considering the question about radiation belt or magnetospheric PsA processes leading to precipitation, they are not addressed in our study but we are focusing on atmospheric impact from PsA-EEP. They have, however, been addressed in recent studies, e.g., by Miyoshi et al. (Geophys. Res. Lett., 47, e90360, doi:10.1029/2020GL090360, 2020) and references therein).

30   2. Recommend authors provide a deeper discussion about the uncertainties associated with the "MCMC median" ionization profile. The authors appear to use an energy spectrum from a single event (17 November 2012) to drive the entire simulation. Turunen et al. suggest large difference in O3 reductions... 10s of percent... depending on energy spectra. Tesema et al. (figure 4) show a large range of possible energy spectra. What observations are needed in order to better constrain this estimate and associated variability? How might the spectra vary with magnetospheric activity, given changing pitch angle distributions and
35   anisotropies in precipitation? What are uncertainties associated with assuming the same PsA-EEP forcing throughout the year given that previous studies such as Bland et al. identify seasonal differences in occurrence rates?

Response: Turunen et al. spectrum is indeed based on a single event, but it is in agreement with the statistical study by Tesema et al. (2020) making it a well-validated median spectrum and is thus well-suited for our long-term impact study. To constrain the spectral variability with magnetic activity, in the future we would need more detailed studies of precipitation spectra during different types of PsA, including the effect of patchiness. Because the PsA-EEP simulation uses a 50% wintertime occurrence frequency that is higher than in summer (Bland et al., 2019), by applying median PsA forcing throughout the year we are overestimating the summertime forcing by a factor of about 2.5 in our simulations. For the assessment PsA impact, however, this has a small overall impact: our results show that the long-term atmospheric response is clearly driven by the wintertime forcing.

In the revised manuscript, we have revised the text on spectrum selection and uncertainties.

3. The authors emphasize the importance of using the full WACCM-D chemistry. It would be helpful to quantify the difference on $NO_x$ production using this chemistry as a function of altitude and electron precipitation energy spectra. That is, at what altitudes and electron energies is using the full WACCM-D chemistry most critical?

Response: As shown by Andersson et al. (2016) in the case of the January 2005 solar proton event, detailed D-region chemistry resulted in 30–130% more $NO_x$ at 70–85 km compared to the standard parameterization. We expect a similar factor-of-two increase in PsA-EEP $NO_x$ response from the detailed ion chemistry. Note that this is direct enhancement in the mesosphere at electron energies of about 40–200 keV and, depending on dynamical conditions, the extra $NO_x$ can then descend and affect lower altitudes.

In the revised manuscript, we have added this information (Section 2).

4. Recommend adding a more thorough discussion of why these seasonal and spatially limited O3 reductions are important in atmospheric processes at various altitudes (dynamics, radiative transfer, chemistry). For example, why is a 5% decrease in O3 within the winter polar vortex at 40 km important (e.g., Figure 7)? And how significant is this decrease with respect to other energetic particle precipitation impacts (i.e., solar protons, GCRs, and other sources of electrons)?

Response: The simulated 5% ozone depletion from PsA-EEP alone is a substantial contribution to the total EPP impact because it is comparable to that seen in satellite observations (up to 15% in the SH upper stratosphere, Damiani et al. 2016) and in simulations (e.g. 7% in the SH upper stratosphere, Andersson et al. 2016). Capturing the magnitude of the stratospheric ozone response is important for realistic simulations of the proposed ground-level climate connection because middle atmospheric ozone controls the dynamical response through absorption of solar ultraviolet radiation. The ozone response to EPP is typically seen in the polar cap areas (as shown e.g. in Figure 7), but these ozone changes affect the temperature balance between the mid and polar latitudes, and subsequently the zonal winds, and connects to the ground-level climate variability (e.g. Baumgaertner et al. 2011). Currently, the EPP-related ground-level regional temperature variability from observations ($\pm5$ K, Seppälä et al. 2009) exceeds the simulated variability ($\pm1$ K, Rozanov et al. 2012), and improvements in the EPP forcing could help to reduce the difference.

In the revised manuscript, we have improved the text in Section 4.

Minor:

Line 38 – "40%-60% ozone depletion" at what altitudes?
Response: In the revised manuscript, "in the polar upper stratosphere" has been added.

Line 79 – 80% at what altitude?

Response: In the revised manuscript, "at 75–80 km" has been added.

75   Line 144 - what electron energies do "below 85 km" correspond to?
Response: In the revised manuscript, "(electron energies larger than about 40 keV)" has been added.

Lines 223-225 – It would be useful to mention this model "spin-up" earlier in the paper. (Or showing results just for the second winter)
Response: In the revised manuscript, we mention this in Section 2. Note that the discussion on the NH results is already
80   focusing on the second winter.

Line 272-273 – Quantify "severely depleted"
Response: In the revised manuscript, this is replaced "by more than 70%".

Lines 305-307 or below – When referencing the limitations of the CMIP MEE electron precipitation estimates, recommend mentioning that APEEP uses the MEPED 0 degree telescope and does not fully take into account pitch angle anisotropies.
85   (Authors reference the work of Nesse Tyssøy et al., but it would be useful to explain more thoroughly in the text).
Response: The underestimation of the CMIP6 MEE which is based on the measurements from the POES/MEPED 0° telescope and makes no use of the 90° telescope could indeed be related to the incomplete pitch angle coverage (e.g. Nesse-Tyssøy et al., 2019). Note, however, that even the use of both telescopes requires assumptions about the pitch angle distribution. In the context of the current paper, we feel that addition of these details would require an extensive discussion of
90   MEPED observations which would be a distraction for the reader. Thus we decided not to add these details, and trust that the reviewer can agree with our view.

Technical corrections:

Line 139 – "patterns" (typo)
Response: In the revised manuscript this is corrected.

---

## Author Comment (AC4)

**Authors' response to reviewers' comments on "Simulated seasonal impact on middle atmospheric ozone from high-energy electron precipitation related to pulsating aurorae" by Verronen et al.**

Please find below our answers (in blue) to the comments (in black).

**Response to the comments of Referee #3 (Allison Jaynes)**

5 General Comments: This manuscript analyzes the seasonal impacts of energetic precipitation from pulsating aurora by integrating realistic energy spectrums and spatial extents of precipitation into WACCM simulations including lower ionospheric chemistry. The results clearly show the descent of $NO_x$ to lower altitudes in the winter, which causes a significant portion of the ozone loss. There are also clear differences between the southern and northern hemispheres, due to variations in the polar vortex. This study is a very nice example of the effect that energetic electrons from pulsating aurora can have on the

10 atmosphere and furthers our understanding of this important topic. I have included some comments and suggestions below for consideration, which I hope can be addressed for the final submission.

Response to general comments: We would like to thank the reviewer for her positive comments and appreciate the time devoted to the evaluation of our paper.

Specific comments:

15 Line 53: Add citation for "with a median duration of about 2–4 hours"

Response: The citation is Tesema et al. (2020), we have added it in the revised manuscript.

Line 56: Add citation for "PsA decays slower than the geomagnetic activity recovers"

Response: The citation is Partamies et al. (2017), we have added it in the revised manuscript.

Lines 192-195 & Figure 2: The difference between electron density in thermo-PsA and no-PsA compared to full-PsA and
20 no-PsA is not clear in Figure 2. Essentially, these sentences are commenting on the clear difference between Figures 2e and 2i, but with the current color scale, that difference is not perceptible. There may be a slight bit of darker green in Figure 2e from 70-80 km (or up to 100 km in the winter) but it is certainly not clear and I had to really look several times and read this section closely to see there was a difference. Is it possible to change the color scale in just these two panels to a monochromatic one or else zoom in on this altitude range to get a better use of the rainbow scale to show the difference?

25 Response: Indeed, the electron density difference below 85 km is not quantitatively clear from Figures 2e and 3e, although qualitatively there is an obvious response down to about 63 km in the full-PsA simulation. The D-region electron density response to PsA-EEP ranges from very small in nighttime (when majority of the negative charge below about 80 km is held by ions) to $10^3 – 10^4$ cm$^{-3}$ increase in daytime. In hourly time resolution, the daytime increases are not very clear from the Figure.

30 Since electron density response is not the main focus of the paper, and the hourly resolution works well for other species, we have decided to keep the figures the same. Nevertheless, we see now that the text was not very clear on "what should be compared with what" (comparing the response in the full-PsA simulation to that in the thermo-PsA simulation means comparison between panels e and i, and this is what we discuss in the text.) Therefore, in the revised manuscript, we have

clarified the text to address the reviewer's comment and now give the magnitude of the D-region electron density response in the text.

Figures 5 & 6: Is there a white trace in panels a and c on Figures 5 and 6? Either the white is not visible due to the mostly white background, or you did not perform the 30-day mean of the black line, which should be noted.

Response: True, in Figures 5 and 6, the panels a and c do not present the 30-day averages because the response is clear from the hourly data. We have corrected the figure captions.

Figure 5: In panels a and c, the cyan line does not look like a 30-day running mean. Granted, we cannot see the full data for difference between thermo-PsA and no-PsA, but it does not look similar (as smooth) to Figs 4a, 4c, 6a, 6c. If there is a reason for this, please comment on it in the text, or point to it if I missed it.

Response: Thank you for pointing this out. Indeed, it is not the 30-day average but the hourly difference. As noted above, we did not include the 30-day averages here because the response is clear from the hourly data. We have corrected the figure caption.

Overall comment: Is it reasonable to estimate a total % difference integrated over time or spatial region or both? I see the utility of showing a figure such as Figure 24, but it doesn't give an indication of over the entire winter, say, how much of a contribution is this input of PsA- EEP. Similarly for the entire NH or SH region. Perhaps an analysis of the difference in integrals of the time series shown in Figures 4, 5, 6. This may give a better overall context for the differences due to PsA-EEP over a season.

Response: Both the temporal and spatial integration could be done in many different ways. Currently we do show altitude-integrated responses and also 30-day averages in Figures 4–6. Because the background concentrations vary considerably (panels a-d in Figures 2 and 3) and the responses at individual altitudes have different temporal characteristics (e.g. panels e-h in Figures 2 and 3) over the year (or over the winter period), we prefer to show the relative differences in Figures 4–6 as 30-day averages rather than integrated over longer time periods.

Grammar and spelling:

Line 29: "atmospheric ionization rates due to precipitation of solar protons,"
Line 167: "In wintertime, the largest concentrations…"
Line 194: "… removal of the ionization…"
Line 205: double "are" in "between the NH and SH are seen…"
Line 229: "In the NH, the peak increase…"
Line 245: remove "of" in "because of less variability in the polar vortex dynamics…"

Response: We have corrected grammar and spelling as suggested.